# The Oral Pheneticillin Absorption Test: An Accurate Method to Identify Patients with Inadequate Oral Pheneticillin Absorption

**DOI:** 10.3390/antibiotics8030119

**Published:** 2019-08-15

**Authors:** Anneke C. Dijkmans, Dinemarie M. Kweekel, Jaap T. van Dissel, Michiel J. van Esdonk, Ingrid M. C. Kamerling, Jacobus Burggraaf

**Affiliations:** 1Centre for Human Drug Research (CHDR), 2333 CL Leiden, The Netherlands; 2Department of Clinical Pharmacy, Leiden University Medical Center (LUMC), 2333 ZA LeidenLeiden, The Netherlands; 3Department of Infectious Diseases, Leiden University Medical Center, 2333 ZA Leiden, The Netherlands

**Keywords:** penicillin, oral, pheneticillin, streptococcal

## Abstract

Severe streptococcal infections are commonly treated with intravenous followed by oral penicillin (pheneticillin) therapy. However, switching from iv to oral therapy is complicated by the variability in oral pheneticillin absorption. We employed an Oral Absorption Test (OAT) for pheneticillin to identify patients in whom oral pheneticillin absorption is poor. Out of 84 patients 30 patients (36%) were identified as insufficient absorbers. Treatment failure due to pheneticillin malabsorption can be avoided by performing an OAT, and these patients should be treated by another antibiotic, which is known to be absorbed well.

## 1. Background

In the Netherlands, patients with severe streptococcal infections are effectively treated with narrow-spectrum antibiotics, most frequently with initial intravenous (iv) penicillin G followed by oral maintenance therapy, usually with pheneticillin. The advantages of this treatment approach are the bactericidal activity of penicillin, the low costs, the possibility to switch to an oral antibiotic as soon as possible, and the lower risk of introducing resistance associated with narrow-spectrum antibiotics. However, the disadvantage is the highly variable absorption of pheneticillin [1], and patients in whom adequate dosing is critical should undergo an oral pheneticillin absorption test before switching to oral pheneticillin maintenance therapy. The purpose of this report is to describe our findings with the oral absorption test (OAT) implemented in our medical center.

## 2. Patients and Methods

The gathering of the data for this investigation complies with Dutch law as the evaluation concerned with daily routine practice; the law on the medical treatment agreement (WGBO; Wet op de Geneeskundige Behandelings Overeenkomst). Hence, separate medical ethical approval was not needed.

The evaluation included patients admitted to Leiden University Medical Centre, Leiden, the Netherlands from 2005 to 2013. Data were collected from hospitalized patients with the only inclusion criterion that they received initial penicillin G and were scheduled for maintenance treatment with oral pheneticillin.

### Pheneticillin Oral Absorption Tests

Like other beta-lactam antibiotics, treatment efficacy of pheneticillin is defined by the time the plasma concentration is above minimum inhibitory concentration (MIC). As a proxy for this, we used pheneticillin peak concentrations as absorption is the limiting factor to reach bactericidal effects after oral administration. While on iv penicillin G therapy, patients received an oral dose of 1 g of pheneticillin in the fasted state. Blood samples for serum pheneticillin concentration were taken at baseline and at 1 and 2 h after the oral dose. These sample times were chosen because of the expected time of maximal concentration at 1 h after intake [1]. However, because the time to maximal concentrations cannot be predicted reliably for individual patients, it was decided to have a relatively wider window and take samples at 1 and 2 h as this would allow assessment of the absorption also in case of diminished gastrointestinal motility. Adequate absorption was defined as an increase of ≥10 mg/L pheneticillin relative to trough concentration (t = 0) either 1 or 2 h after dosing. The reason for choosing this concentration is that the highest MIC value indicating susceptibility is defined as <0.5 mg/L of free drug (e.g., the breakpoint MIC for most microorganisms for phenoxypenicillins by EUCAST [2]). Given the high protein binding (80%) [1] of pheneticillin, total serum drug concentration should be at least 5 mg/L. We used a safety margin and considered serum pheneticillin concentrations above 10 mg/L as therapeutic levels. The maximal pheneticillin level during the absorption test (at either 1 or 2 h after dosing) was taken forward in the analysis to decide on adequate absorption of a patient. An unpaired t-test was applied to explore sex differences in oral absorption of pheneticillin. Correlations and the adjusted R^2^ of linear regression were calculated to identify any trends of oral absorption over age. The effect of diabetes mellitus or use of gastric acid inhibitors on pheneticillin absorption was also investigated.

## 3. Pheneticillin Assay

Pheneticillin concentration was determined with high performance liquid chromatography [3]. This method allowed for the simultaneous detection and quantification of pheneticillin, benzylpenicillin, and flucloxacillin in a single sample. The assay shows linearity for pheneticillin concentrations up to 50 mg/L with a lower limit of quantification of 3 mg/L. The accuracy and reproducibility of the method (determined by repeated measurement of a quality control sample) were 103.3 and 5.6%, respectively [3].

## 4. Results

Eighty-four (84) hospitalized patients (59 males, 25 females), mean age ± standard deviation (range) = 58.3 ± 15.4 (19–90 years), were included in the analysis. A total of 30 patients (36%) were identified as insufficient absorbers. The median increase from baseline was 11.6 mg/L (inter-quartile range = 8.5–16.1 mg/L, range = 3.8–27.7 mg/L) as depicted in Figure 1. The majority of maximal concentrations were reached at 1 h after dosing (80%).

No linear correlation between the maximal increase from baseline and age was detected (correlation 0.22, adjusted R^2^ = 0.04) (Figure 2A). No significant difference between gender in maximal absorption was identified (*p* = 0.66). A wide scatter in the maximal absorption between patients having diabetes mellitus and/or on gastric acid inhibitors was observed (Figure 2B); no formal statistical analysis was performed due to the low sample sizes.

## 5. Conclusions and Discussion

There is little doubt that the use of narrow-spectrum antibiotics is effective to not further aggravate the increasing problem associated with of multi-resistant bacterial resistance. Severe streptococcal infections are commonly adequately treated with intravenous penicillin followed by oral therapy. Indeed, there are many advantages to start oral therapy as soon as possible. However, the early switch from penicillin G to pheneticillin is complicated by the variability in oral pheneticillin absorption, which may jeopardize treatment outcome, especially in cases in which adequate dosing is pivotal. Future research should investigate additional variables that explain the observed variability in order to optimize dose selection in this heterogeneous patient population. Treatment failure due to pheneticillin malabsorption can be simply avoided by performing an OAT. In case it is not feasible to perform an OAT, knowing the high rate of non-absorbers, we advise not to switch to oral pheneticillin and choose for another antibiotic, such as—if feasible—amoxicillin per os, which is known to be absorbed well [1,4,5].

## Figures and Tables

**Figure 1 antibiotics-08-00119-f001:**
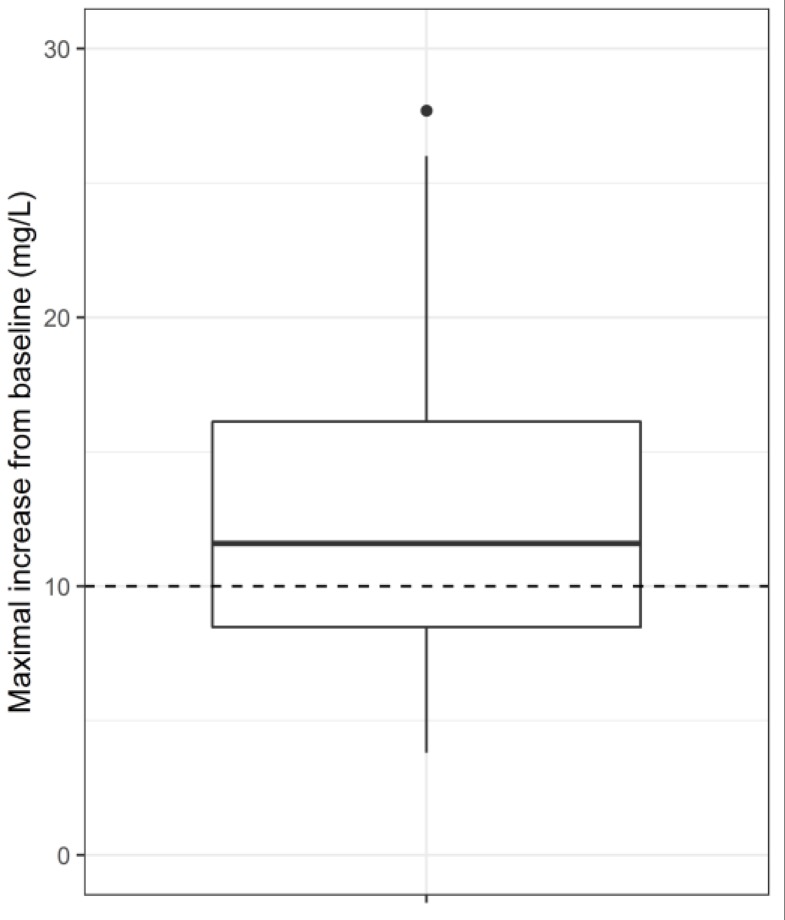
Distribution of maximal increase from baseline in pheneticillin showing the median, inter-quartile range, and outliers. Horizontal dashed line indicates the cut-off for adequate absorption (10 mg/L).

**Figure 2 antibiotics-08-00119-f002:**
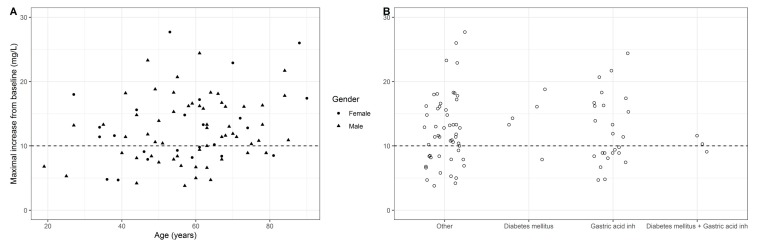
(**A**) Maximal increase from baseline versus age for men and women. (**B**) Maximal increase from baseline for diabetes mellitus patients and/or patients on gastric acid inhibitors. Horizontal dashed line indicates the cut-off for adequate absorption (10 mg/L).

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
