# Peer review of "The Oral Pheneticillin Absorption Test: An Accurate Method to Identify Patients with Inadequate Oral Pheneticillin Absorption"

_antibiotics, 2019, doi:10.3390/antibiotics8030119_

Round 1

Reviewer 1 Report

In their brief report, Dijkmans and colleagues provide a short description of pheneticillin absorption in 88 hospitalized patients receiving iv penicillin G for streptococcal infections, with subsequent step-down to oral pheneticillin. Overall, the letter is well written. I have only a few minor remarks. 

Minor comments

1) The authors did not show data regarding co-morbidity, co-medication, and renal function. Although these factors were not associated (which statistical test was used for evaluating possible associations?) with the insufficient absorber status, they may be of interests for future comparisons with other cohorts and they can be added as supplementary tables/figures.

2)    Please specify also in methods that patients receive the treatment for streptococcal infections

3)    Lines 43. 19-90 years. Probably a range and not an IQR, but it should be specified.

Author Response

L.S.

Thank you very much for reviewing our manuscript. We will explain point-by-point the details of the
revisions in the manuscript and our responses to the your comments.
Please see below.

Kind regards, Anneke Dijkmans

Minor comments

1) The authors did not show data regarding co-morbidity, co-medication, and renal function. Although these factors were not associated (which statistical test was used for evaluating possible associations?) with the insufficient absorber status, they may be of interests for future comparisons with other cohorts and they can be added as supplementary tables/figures. You do have a good point. We rewrote this and we provided far more information in the manuscript.

2)    Please specify also in methods that patients receive the treatment for streptococcal infections Actually it was mostly for streptococcal infections, but also for Proprioni, Actinomyces and sometimes even for S.aureus infections.

3)    Lines 43. 19-90 years. Probably a range and not an IQR, but it should be specified. We specified this.

Reviewer 2 Report

Brief Summary:

The brief report by Dijkmans and colleagues provides insight into one institution’s pheneticillin pharmacokinetic findings and therefore has the potential to make a meaningful clinical contribution to the medical literature surrounding pheneticillin use. The largest limitations reside within the results section with limited data being presented.

Suggested Modifications:

On page 1 lines 2-3, the title appears to convey that the intention of the article will be to discuss the feasibility of implementation of an oral pheneticillin absorption test. Consider rewording. On page 1 line 14 and page 2 line 53, there is reference to “small spectrum antibiotics.” This phrasing is inconsistent with commonly accepted nomenclature. I would recommend rephrasing as “narrow-spectrum antibiotics” to maintain consistency.

On page 1 lines 19-20, it is mentioned that an oral pheneticillin test may be beneficial in patients for whom adequate dosing is critical, however there is no mention of a specific subpopulation in which it is essential to achieve certain concentrations or particular variables which may have a considerable impact on pheneticillin absorption. It would be beneficial to include a statement regarding which populations are at increased risk for malabsorption or if there has been evidence behind clinical or microbiological failure with pheneticillin malabsorption.

On page 1 line 36, citation #2 is referenced in regards to expected maximal concentration of pheneticillin. However, the cited article only discusses amoxycillin and ampicillin with no mention of pheneticillin. Citation #1 contains information regarding phenethicillin concentrations in healthy subjects over specified time points and may be used to explain time points selected.  In addition, per these statements the authors have selected approximate peak concentrations as a surrogate for anticipated maintenance of appropriate serum concentrations throughout therapy. Pheneticillin is a beta-lactam antibiotic which are optimized through maximizing the time spent above the minimum inhibitory concentration (MIC). The typical surrogate marker used to estimate the time spent above the MIC are trough values and not peak concentration values. If possible, the authors should provide a statement regarding why peak concentrations were examined in this study instead of other pharmacokinetic values.

On page 1 line 39, a threshold of ≥ 10 mg/dL is specified to classify adequate absorbers, however, no citation is provided. At the very least a citation is required regarding where this threshold was derived and the clinical implications of this threshold would be beneficial.

On page 2 line 41, citation #3 is referenced in regards to determining pheneticillin concentration with high performance liquid chromatography (HPLC). Upon review of the cited article, there is no mention of pheneticillin or utilization of HPLC to quantify pheneticillin concentrations nor does the cited article validate HPLC use in quantification of drug concentrations. Please clarify.

On page 2 lines 44-45, outcome results are reported as median increase in concentration; however, no information if given to identify relative initial concentration thus these results are susceptible to misinterpretation (i.e. may be interpreted as an increase from 0 mg/L as was the concentration prior to dose OR as an increase between serum concentration at 1 hour vs. serum concentration at 2 hours post-dose). I would recommend either rephrasing results to convey median concentrations at specified time points or include phrasing which clarifies the relative initial concentration (e.g. “the median increase from time 0 to 1 hour was 11.5 mg/dL).”

For figure 1, similar to the statement regarding page 2 lines 44-45, would suggest reexamining terminology used on the axes, particularly “maximal increase” as this may be susceptible to misinterpretation. It would be acceptable to clarify in the figure description below to help readers understand the information being conveyed.

For figure 2, the variable examined does not have a high level of clinical significance (concentration vs. age). As there was only brief statement regarding the absence of trends with other variables (seen on page 2 lines 46-47), I would recommend displaying a figure that contained a higher level of clinical significance i.e. pheneticillin concentration compared to a variable that has been shown to be closely correlated with variable drug absorption, for example stratified renal dysfunction if applicable based on previous literature.  Also with this figure, were the concentrations obtained all from a 1hr sample or was there a mix of hr and 2 hr samples? This should be clarified.

Overall Recommendation:

Requires considerable revision with comments above. Many are in line with further expanding on the methods of the study and the results in order for another group to recreate this study. 

Author Response

L.S.

Thank you very much for reviewing our manuscript. We will explain point-by-point the details of the
revisions in the manuscript and our responses to the your comments.
Please see below. We think this significantly improved the manuscript.

Kind regards, Anneke Dijkmans

On page 1 lines 2-3, the title appears to convey that the intention of the article will be to discuss the feasibility of implementation of an oral pheneticillin absorption test. Consider rewording.We changed the title into The oral pheneticillin absorption test: an accurate method to identify patients with inadequeate oral pheneticillin absoprtion On page 1 line 14 and page 2 line 53, there is reference to “small spectrum antibiotics.” This phrasing is inconsistent with commonly accepted nomenclature. I would recommend rephrasing as “narrow-spectrum antibiotics” to maintain consistency. Thank you, this is done.

On page 1 lines 19-20, it is mentioned that an oral pheneticillin test may be beneficial in patients for whom adequate dosing is critical, however there is no mention of a specific subpopulation in which it is essential to achieve certain concentrations or particular variables which may have a considerable impact on pheneticillin absorption. It would be beneficial to include a statement regarding which populations are at increased risk for malabsorption or if there has been evidence behind clinical or microbiological failure with pheneticillin malabsorption.All patients suffered from severe infections and started with penicillin G. After intravenous treatment, oral maintainance therapy was needed, therefore absorption of pheneticillin is required. In the case of malabsorption amoxycillin (or clindamycin) are advised.

On page 1 line 36, citation #2 is referenced in regards to expected maximal concentration of pheneticillin. However, the cited article only discusses amoxycillin and ampicillin with no mention of pheneticillin. Citation #1 contains information regarding phenethicillin concentrations in healthy subjects over specified time points and may be used to explain time points selected. Thank you, this was a mistake, we changed #2 in 1#

In addition, per these statements the authors have selected approximate peak concentrations as a surrogate for anticipated maintenance of appropriate serum concentrations throughout therapy. Pheneticillin is a beta-lactam antibiotic which are optimized through maximizing the time spent above the minimum inhibitory concentration (MIC). The typical surrogate marker used to estimate the time spent above the MIC are trough values and not peak concentration values. If possible, the authors should provide a statement regarding why peak concentrations were examined in this study instead of other pharmacokinetic values.  Yes, that is correct. But we need to know if patients absorbs well. We reworded the text:

Like other beta-lactam antibiotics, treatment efficacy of pheneticillin is defined by the time the plasma concentration is above minimum inhibitory concentration (MIC). As a proxy for this we used pheneticillin peak concentrations as absorption is the limiting factor to reach bactericidal effects after oral administration. While on iv penicillin G therapy, patients received an oral dose of 1 gram of pheneticillin in the fasted state. Blood samples for serum pheneticillin concentration were taken at baseline and at 1 and 2 hrs after the oral dose. These sample times were chosen because of the expected time of maximal concentration at 1 hr after intake(1). However, because the time to maximal concentrations cannot be predicted reliably for individual patients, it was decided to have a relative wide window and take samples at 1 and 2 hrs as this would allow assessment of the absorption also in case of diminished gastrointestinal motility. Adequate absorption was defined as an increase of ≥ 10 mg/L pheneticillin relative to trough concentration (t=0) either 1 or 2 hours after dosing.. The reason for choosing this concentration is that the highest MIC value indicating susceptibility, is defined as <0.5 mg/L of free drug (eg the breakpoint MIC for most micro-organisms for phenoxypenicillins by EUCAST(2)). Given the high protein binding (80%)(1) of pheneticillin total serum drug concentration should be at least 5 mg/L. We used a safety margin and considered serum pheneticillin concentrations above 10 mg/L as therapeutic levels. The maximal pheneticillin level during the absorption test (at either 1 or 2 hrs after dosing) was taken forward in the analysis to decide on adequate absorption of a patient. An unpaired t-test was applied to explore sex differences in oral absorption of pheneticillin. Correlations and the adjusted R2 of linear regression were calculated to identify any trends of oral absorption over age. The effect of diabetes mellitus or use of gastric acid inhibitors on pheneticillin absorption was also investigated.

On page 1 line 39, a threshold of ≥ 10 mg/dL is specified to classify adequate absorbers, however, no citation is provided. At the very least a citation is required regarding where this threshold was derived and the clinical implications of this threshold would be beneficial. We explained it in the text. Unfortunately no citation is available. We use this for decades in our institution, for example in Therapeutic Drug Monitoring for flucloxacillin (https://www.ncbi.nlm.nih.gov/pubmed/22569354) or in Clinical Infectious Diseases   https://academic.oup.com/cid/article/17/3/491/363061

On page 2 line 41, citation #3 is referenced in regards to determining pheneticillin concentration with high performance liquid chromatography (HPLC). Upon review of the cited article, there is no mention of pheneticillin or utilization of HPLC to quantify pheneticillin concentrations nor does the cited article validate HPLC use in quantification of drug concentrations. Please clarify. This is the same assay. Please see the text:

Pheneticillin concentration was determined with high performance liquid chromatography(3). This method allowed for the simultaneous detection and quantification of pheneticillin, benzylpenicillin and flucloxacillin in a single sample. The assay shows linearity for pheneticillin concentrations up to 50 mg/L with a lower limit of quantification of 3 mg/L. The accuracy and reproducibility of the method (determined by repeated measurement of a quality control sample) were 103.3% and 5.6%, respectively(3).

On page 2 lines 44-45, outcome results are reported as median increase in concentration; however, no information if given to identify relative initial concentration thus these results are susceptible to misinterpretation (i.e. may be interpreted as an increase from 0 mg/L as was the concentration prior to dose OR as an increase between serum concentration at 1 hour vs. serum concentration at 2 hours post-dose). I would recommend either rephrasing results to convey median concentrations at specified time points or include phrasing which clarifies the relative initial concentration (e.g. “the median increase from time 0 to 1 hour was 11.5 mg/dL).” This is corrected in the text. We included baseline.

For figure 1, similar to the statement regarding page 2 lines 44-45, would suggest reexamining terminology used on the axes, particularly “maximal increase” as this may be susceptible to misinterpretation. It would be acceptable to clarify in the figure description below to help readers understand the information being conveyed. This is corrected. Thank you.

For figure 2, the variable examined does not have a high level of clinical significance (concentration vs. age). As there was only brief statement regarding the absence of trends with other variables (seen on page 2 lines 46-47), I would recommend displaying a figure that contained a higher level of clinical significance i.e. pheneticillin concentration compared to a variable that has been shown to be closely correlated with variable drug absorption, for example stratified renal dysfunction if applicable based on previous literature.  Also with this figure, were the concentrations obtained all from a 1hr sample or was there a mix of hr and 2 hr samples? This should be clarified. We specified information about 1 and 2 hours samples in the manuscript and we specified absorption in patients with DM and gastric acid inhibitors.